# CSPG4 as Target for CAR-T-Cell Therapy of Various Tumor Entities–Merits and Challenges

**DOI:** 10.3390/ijms20235942

**Published:** 2019-11-26

**Authors:** Dennis C. Harrer, Jan Dörrie, Niels Schaft

**Affiliations:** Department of Dermatology, Universtitätsklinikum Erlangen, Friedrich-Alexander-Universität Erlangen-Nürnberg, Hartmannstraße 14, 91052 Erlangen, Germanyjan.doerrie@uk-erlangen.de (J.D.)

**Keywords:** CSPG4, target antigen, CAR-T cell, melanoma, leukemia, glioblastoma, triple-negative breast cancer

## Abstract

Targeting cancer cells using chimeric-antigen-receptor (CAR-)T cells has propelled adoptive T-cell therapy (ATT) to the next level. A plentitude of durable complete responses using CD19-specific CAR-T cells in patients suffering from various lymphoid malignancies resulted in the approval by the food and drug administration (FDA) of CD19-directed CAR-T cells for the treatment of acute lymphoblastic leukemia (ALL) and diffuse large B-cell lymphoma (DLBCL). A substantial portion of this success in hematological malignancies can be traced back to the beneficial properties of the target antigen CD19, which combines a universal presence on target cells with no detectable expression on indispensable host cells. Hence, to replicate response rates achieved in ALL and DLBCL in the realm of solid tumors, where ideal target antigens are scant and CAR-T cells are still lagging behind expectations, the quest for appropriate target antigens represents a crucial task to expedite the next steps in the evolution of CAR-T-cell therapy. In this review, we want to highlight the potential of chondroitin sulfate proteoglycan 4 (CSPG4) as a CAR-target antigen for a variety of different cancer entities. In particular, we discuss merits and challenges associated with CSPG4-CAR-T cells for the ATT of melanoma, leukemia, glioblastoma, and triple-negative breast cancer.

## 1. Introduction

T cells redirected to malignant cells via chimeric antigen receptors (CARs) have induced spectacular responses in patients suffering from relapsed and refractory hematological malignancies [1,2,3]. Predicated on numerous complete responses in leukemia and lymphoma patients achieved via a single infusion of genetically engineered CAR-T cells, official approval was recently issued by the food and drug administration (FDA) as well as by the European medicines agency (EMA) for the use of CD19-CAR-T cells in acute lymphoblastic leukemia (ALL) and diffuse large cell B-cell lymphoma (DLBCL) [4]. 

CARs are created by assembling an antibody-derived single chain Fv (scFv) and the intracellular part of the CD3ζ chain linked in cis with a co-stimulatory domain [5]. This modular composition allows for T-cell activation in response to antigens located on the surface of malignant cells by binding of the single chain Fv and subsequent signaling through the CD3ζ chain and the co-stimulatory domain [5]. Co-stimulation is mostly provided by either the immunoglobulin superfamily member CD28 or 4-1BB pertaining to the tumor necrosis factor (TNF) receptor superfamily [5]. Whereas CD28 activity polarizes T cells towards effector cells relying on glycolytic energy metabolism and evincing profound effector functions at the expense of a limited persistence, 4-1BB imposes a shift towards fatty acid oxidation and memory destiny, resulting in enhanced longevity of 4-1BB co-stimulated CAR-T cells [6]. 

An easy but unfortunately very effective process by which tumor cells can escape recognition by CAR-T cells is antigen down-regulation or antigen-loss [7]. Hence, it is crucial to establish a comprehensive repository of backup targets to prepare for antigen shutdown. Conspicuously, the success of CAR-T-cell therapy is by now confined to hematological malignancies. Attacking solid tumors with CAR-T cells entails some additional impediments, such as the need to survive and display effector functions inside a harsh tumor microenvironment (TME) with restricted access to nutrients and an abundance of immunosuppressive molecules (reviewed in [8,9]). Transforming growth factor β (TGFβ), for instance, is highly active in repressing CAR-T-cell effector functions by both directly impeding T-cell activation and by reprogramming effector T cells into tumor-protective regulatory T cells [10]. Another immunosuppressive cytokine in the TME is interleukin (IL-)10, which blocks the activation of cytotoxic killer cells and natural killer cells [11]. Cellular components of the TME protecting cancer cells from T-cell-mediated immunity include regulatory T cells, myeloid derived suppressor cells and tumor-associated macrophages [11]. Regulatory T cells secrete large quantities of the immunosuppressive cytokines, TGFβ and IL-10 [11]. Myeloid-derived suppressor cells deplete arginine via the enzyme arginase leading to impaired T-cell proliferation in the TME [11]. Tumor-associated macrophages constitute a major source of IL-10 in the TME resulting in reduced T cell activation [11]. Moreover, CAR-T cells exhibit limited persistence in solid tumors [12]. One of the most severe issues, however, arises from the paucity of suitable target antigens in solid tumors. Ideal targets unify three essential attributes (Figure 1): i) uniform presence on the surface of malignant cells reducing the risk for antigen-negative escape variants; ii) absent expression on non-malignant host cells precluding on-target/off-tumor activity, which harbors the potential for severe, potentially lethal, side-effects [13]; and iii) crucial role as an oncogenic driver in cancer cells, which may compound antigen-shutdown due to the selective survival advantage conferred on malignant cells. Antigens that are not instrumental in oncogenesis, such as CD19 in ALL and DLBCL are prone to shutdown [7]. Co-expression on by-stander cells maintaining the tumor-microenvironment, such as tumor-associated vasculature, fibroblasts and macrophages represents another beneficial trait. Taken together, the quest for appropriate CAR-target remains one of the most important tasks to pave the way for a successful CAR-T-cell therapy in solid tumors. In this review, we focus on chondroitin sulfate proteoglycan 4 (CSPG4) as a promising target antigen for CAR-T-cell therapy.

CSPG4, formerly termed melanoma-associated chondroitin sulfate proteoglycan (MCSP) or high molecular weight melanoma-associated antigen (HMW-MAA) [14,15], is a type 1 single pass transmembrane protein encoded by chromosome 15 [16]. Of note, neural/glial antigen 2 (NG2) represents the rat ortholog with approximately 81% amino acid sequence homology [17]. Usually, the protein core of CSPG4 is subjected to heavy glycosylation in the Golgi apparatus before being translocated to the plasma membrane [18]. Additionally, the glycoprotein backbone incorporates three glycosaminoglycan (GAG) sites for chondroitin sulfate (CS) attachment giving rise to a 450kDa proteoglycan [19]. As a result of the myriad glycosylation patterns and the variability in CS anchorage, CSPG4 appears in different glycoforms, which vary according to cell type or differentiation status. The large extracellular part is subdivided into three distinct domains. The membrane-proximal domain D3 is a glycosylated globular binding site for lectins, such as galectin-3 [20] and P-selectin [21], as well as for integrin α1β3 [20]. The large D2 domain is composed of 15 CSPG4-specific repeats and harbors CS attachments sites, integrin-binding sites, collagen-binding sites and it is supposed to act as a growth factor repository, presenting those to adjacent receptor tyrosine kinases (RTK) [18]. The *N*-terminal disulfide-bond-stabilized D1 part contains two laminin G-type domains, which are involved in interactions with the extracellular matrix [18]. The short intracellular domain, which does not possess intrinsic catalytic capacities itself, features two threonine phosphorylation sites, Thr2256 for protein kinase C alpha (PKCα) and Thr2314 for extracellular signal related kinase [22]. The C-terminally located PDZ domain creates docking sites for adaptor molecules, such as multi-PDZ domain protein 1 (MUPP1) and syntenin-1, which, possibly supported by a proline-rich motif, establish a connection to the actin cytoskeleton through protein scaffold formation [23,24]. Direct association of extracellular signal-regulated protein kinases 1 and 2 (ERK1/2) with CSPG4 via a D-domain has also been confirmed [18]. Two major signaling pathways ramify from the intracellular part of CSPG4. The first is the mitogen-activated-protein-kinase (MAPK) pathway either directly triggered by ERK1/2 activation or indirectly via adjacent RTKs stimulated by CSPG4-bound growth factors, such as platelet-derived growth factor (PDGF) [22,25]. Secondly, the interactions with integrins and the actin cytoskeleton forge a link to the focal adhesion kinase (FAK) pathway responsible for sensing changes in the extracellular matrix [26]. Importantly, phosphoinositide 3-kinase (PI3K) and AKT1 are also downstream mediators of CSPG4 signaling [27]. Although the precise physiological function of CSPG4 requires further elucidation, it has been implicated in promoting proliferation and survival, mediated by MAPK signaling and cross-presentation of growth factors [28]. Furthermore, based on the interconnection with the actin cytoskeleton and the binding to various integrins and components of the extracellular matrix, CSPG4 plays a role in cell motility and tissue invasion [29]. Finally, CSPG4 has been implicated in placenta formation [30], angiogenesis [20], neuronal network formation [31], keratinocyte turnover, and epidermal stem cell homeostasis [32]. In sum, CSPG4 is thought to govern various developmental and homeostatic processes across multiple different cell types. At the RNA level, CSPG4 displays a broad distribution throughout normal tissues, such as skin, smooth muscle, pericytes, adipose tissue, uterus, prostate, central nervous system, lung, heart, lymphoid organs, and intestinal tissue [33]. At the protein level, nonetheless, CSPG4 expression seems to be confined to the small intestine [34]. Moreover, CSPG4 displays a broad expression across several different cancer entities, such as melanoma [35], leukemia [36], glioma [37], triple-negative breast cancer [38], head and neck cancers [39], and mesenchymal cancers [40]. In the present review, we want to shine the light on CSPG4 as a promising target antigen for CAR-T-cell therapy. We briefly outline the status quo regarding CAR-T-cell therapy of melanoma, leukemia, glioblastoma, and triple-negative breast cancer. Subsequently, we discuss merits and challenges that CSPG4-CAR-T cells may entail on the ATT of those ill-fated malignancies. 

## 2. CSPG4-CAR-T-Cell Therapy of Different Tumors

### 2.1. Melanoma

Although still eclipsed by checkpoint blockade and kinase inhibitors, preclinical studies and clinical trials on CAR-T cells targeting melanoma have been steadily accumulating. Recently, promising data on HER2-specific CAR-T cells eradicating cutaneous and uveal melanoma cells in humanized mouse models were shared by investigators from Sweden [41]. Remarkably, those CAR-T cells were even able to kill melanoma cells from patients resistant to autologous tumor infiltrating lymphocyte (TIL) infusion therapy [41]. This study underscores the potential of HER2 as a melanoma target antigen and encourages the use of CAR-T cells in highly refractory disease. The integrin αvβ3 represents another emerging target antigen in melanoma, as testified to by long-term tumor-free survival and complete melanoma regression in a xenograft murine model via a single infusion of αvβ3-specific CAR-T cells [42]. Additionally, ganglioside GD3 [43] and CD20 [44] have been successfully exploited as CAR target structures on melanoma cells. Moreover, several clinical trials evaluating the safety and efficacy of CAR-T cells directed against c-MET (NCT03060356), CD20 (NCT03893019), and CD70 (NCT02830724) are recruiting patients. Efforts to impede tumor growth by destroying tumor-associated vasculature using vascular endothelial growth factor receptor 2 (VEGFR2)-specific CAR-T cells conferred dramatic survival advantages on melanoma-bearing mice [45]. Importantly, VEGFR2 is expressed by endothelial cells in the TME and not by melanoma cells per se [45]. This lack of directly targeting melanoma cells might help to rationalize the sobering results observed in melanoma patients treated with VEGFR2-specific CAR-T cells in conjunction with IL-2 (NCT01218867). In this trial, most patients showed progressive disease and only a single patient displayed a partial response. Ganglioside GD2 constitutes another well characterized antigen on several different tumors including melanoma. CAR-T cells reprogrammed to GD2 lyse primary melanoma cells in vitro and cause melanoma elimination in animal models [46]. Data from a previous clinical trial analyzing GD2-specific CAR-T cells are still pending (NCT02107963). In the following we will discuss the merits and challenges of CSPG4 as a target for the CAR-T-cell therapy of melanoma (Figure 2).

#### 2.1.1. CSPG4-CAR-T cells: Merits (Melanoma)

Uniform presence of the target antigen on primary tumor cells as well as on metastases represents an essential precondition for achieving complete remissions. CSPG4 is expressed in the majority of melanoma cells, and owing to its distinct role in promoting metastasis formation and cell motility, it is usually preserved on the surface of melanoma metastases [28,35]. Hence, primary melanoma lesions as well as secondary metastatic lesions are susceptible to CSPG4-CAR-T cell therapy. By contrast, HER2 expression is only detected in a minority of primary cutaneous lesions and recurrent metastatic lesions, as established by immunohistochemical staining of 600 specimens derived from primary or cutaneous lesions or metastases [47]. The presence of CD70 in melanoma cells exhibits a dichotomy between uniform expression on primary lesions and a considerably lower expression in metastases, which predisposes the selection of antigen-negative clones compromising tumor eradication [48]. CD20 is only found on a small percentage of melanoma cells, which are implicated in maintaining the melanoma stem cell pool [49]. Indeed, a small number of studies reported dramatic melanoma regression, ensuing anti-CD20 targeting [44,50]. Nevertheless, the limited expression of CD20 on bulk melanoma cells raises concerns about complete regression and relapse. 

Apart from uniform expression, sustained presence of the target antigen throughout the course of CAR-T-cell therapy is equally important for durable tumor control. Antigen-loss during therapy can be obstructed if the target structure is an oncogenic driver promoting survival and growth of cancer cells. Several lines of evidence, which have accumulated over recent years, suggest that CSPG4 fosters melanoma evolution by procuring growth signals and anti-apoptotic signals [28]. In addition, CSPG4 is instrumental in invasion and metastasis formation [18]. Mechanistically, melanoma growth is supported by ERK signaling downstream of CSPG4 and subsequent activation of the MAPK-pathway, which directs proliferation and counteracts pro-apoptotic signals [22]. Furthermore, CSPG4 is supposed to indirectly stimulate MAPK-signaling by using the large extracellular domain to bind and juxtapose growth factors, such as PDGF and basic fibroblast growth factor (bFGF), to adjacent RTKs resulting in enhanced survival and proliferation of malignant cells [18]. Binding of CSPG4 to collagen VI has also been implicated in relaying survival signals to cancer cells by virtue of PI3K activity [51]. Moreover, the interaction of CSPG4 with the extracellular matrix of the tumor microenvironment turns on FAK signaling, resulting in integrin rearrangement and recruitment of cell division control protein 42 homolog (CDC42), activated CDC42 kinase 1 (ACK-1), and p130cas leading to remodeling of the actin cytoskeleton [26,52]. This facilitates cellular motility and invasion, which are the first steps required for spawning metastases. Besides the promotion of growth, dissolution of cohesive cell layers has also been ascribed to CSPG4-induced ERK signaling [53]. By affecting matrix metalloproteinase (MMP)-1-mediated degradation of collagen I, CSPG4 plays an important role in enabling cell migration through the extracellular matrix [54]. Finally, CSPG4 is an important factor in orchestrating tumor neovascularization, which is crucial for the supply with oxygen and nutrients, but also for dispatching metastases to other sites, such as the liver [33]. The repercussions of CSPG4 blockade or down-regulation on melanoma progression can be deduced from experiments blocking the expression or activity of CSPG4. Wang X. et al. reported inhibition of melanoma growth and migration after blocking CSPG4 with a scFv, which was caused by decreased ERK and FAK activity [34]. Correspondingly, sh-RNA-mediated abrogation of CSPG4 expression in a subcutaneous model of A375M melanoma impaired melanoma proliferation and increased the percentage of apoptotic and necrotic tumor cells [33]. Taken together, CSPG4 expedites melanoma progression in myriad ways and thus, CSPG4 may be less prone to antigen-loss during CAR-T-cell therapy. 

Combination therapy represents a promising strategy to put more pressure on cancer cells and to counteract antigen-loss. Yu et al. presented in-vitro results on the simultaneous blockade of CSPG4 and V600E-mutated BRAF using mAb 225.28 together with a selective BRAF inhibitor [55]. This combinational approach outperformed either agent alone in slowing down melanoma growth [55]. To evaluate the feasibility of combining CSPG4-CAR-T cells with BRAF and MEK inhibitors (BRAFi/MEKi), which are part of the standard therapy of melanoma, our group assessed several canonical effector functions of CSPG4-CAR-T cells in the presence of clinically-approved BRAFi/MEKi combinations [56]. Dabrafenib+Trametinib emerged as the most appropriate partner for CSPG4-CAR-T cells, as the cytolytic capacity was not impaired and cytokine secretion was only moderately reduced [56]. In another study using GD2-CARs, Dabrafenib was also found to be the kinase inhibitor least detrimental to CAR-T-cell activity [46]. Collectively, targeting melanoma with CSPG4-CAR-T cells in conjunction with BRAFi/MEKi might reveal synergistic effects on tumor regression and reduce the frequency of CSPG4 down-regulation by simultaneously engaging two important oncogenic drivers.

#### 2.1.2. CSPG4-CAR-T Cells: Challenges (Melanoma) 

The occurrence of inadvertent on-target/off-tumor toxicity poses one of the most feared side-effects in CAR-T-cell therapy [13]. Hence, scrutinizing the expression profiles of target antigens is an absolute necessity. Using nanostring digital RNA counting, investigators from the national cancer institute (NCI) could confirm the presence of CSPG4 mRNA in a plethora of normal tissues, but at far lower levels than in malignant tissues (ratio tumors: normal tissues: 6.6) [57]. At the protein level, however, CSPG4 exhibited a stricter restriction to cancer cells. Out of 30 normal tissues, including brain, heart, skin, and pulmonary tract, immunohistochemical staining for CSPG4 was only positive in specimen from the small intestine [58]. Furthermore, CSPG4-CAR-T cells did not display cytotoxicity against primary epithelial cell lines from lung, kidney and prostate [59]. Regarding CSPG4 presence on pericytes, it could be demonstrated that CSPG4 is significantly higher expressed in tumor-associated activated pericytes as compared to normal resting pericytes [60]. Nevertheless, on-target/off-tumor toxicity exerted by CSPG4-CAR-T cells against pericytes and other non-malignant tissues stirs serious concerns and poses a major challenge to CSPG4-CAR-T cell therapy. Seeking to curb the potential for on-target/off-tumor toxicity, our group has refined the generation of CSPG4-CAR-T cells via mRNA electroporation [61,62]. Due to the transient CAR expression, arising toxicity will be transient as well. In the clinical setting, it would be reasonable to start with serial mRNA-CSPG4-CAR-T cell infusions, and in the case of absent toxicity proceed to stably transfected CAR-T cells. Other strategies to prevent on-target/off-tumor toxicities encompass the combination of different CARs with different specificities within one T cell. The combination of a first generation CAR (i.e., a CAR without co-stimulatory signaling domains) with chimeric co-receptors (CCR) mediates the necessity for both antigens on the same cell because full T-cell activation is dependent on the combinatorial engagement of both the CAR and CCR in an antigen-specific fashion, which enhances the specificity of the CAR-T cell attack [63]. By using inhibitory CARs (iCARs), one may protect a critical cell population from on-target/off-tumor toxicity. The incorporation of inhibitory signals in a co-expressed CAR can divert T cells off from non-malignant tissues by inhibitory signaling through the iCAR upon binding to antigens over-expressed on certain healthy cells [64]. Which antigen specificities could be combined depends very much on the tumor entity and the health tissue, which should be protected. To control and, if needed, terminate CAR-activity, one can use “ON-switch” CARs. These are characterized by separate antigen-binding and signaling domains, which dimerize upon administration of a certain small molecule allowing for titratable remote control of CAR-T cells with the possibility to decrease the activation level of CAR-T cells in case of toxicities [65]. Furthermore, other groups use SynNotch-induced expression [66] or UniCARs [67] to increase the safety of CAR-T cells. For an overview of all these strategies, please see [68] and [69]. Of note, the challenge to deal with potential of on-target/off-tumor toxicity is not confined to melanoma but bears relevance for CSPG4-CAR-T-cell therapy in general, irrespective of the cancer entity. Another challenge for CSPG4-CAR-T-cell therapy of melanoma, derives from the presence of soluble CSPG4 in the bloodstream [70], which is generated by cleavage of the ectodomain of CSPG4 [31]. Apart from compromising the targetability of melanoma cells by antigen-shedding, soluble CSPG4 might act as a decoy through blocking the CSPG4-binding site of circulating CSPG4-CAR-T cells, which could prevent the recognition of bona fide melanoma cells.

### 2.2. Leukemia

Upon reports from several different groups on spectacular complete remission rates in patients with ALL using CD19-CAR-T cells, a multi-center phase II clinical trial was launched to assess the efficacy of tisagenlecleucel in refractory and relapsing (r/r) ALL [4]. Tisagenlecleucel is a CAR-T-cell product generated by transducing autologous T cells with a 4-1BB-co-stimulated CD19-specific CAR [3]. In this trial, termed ELIANA, 75 patients with ALL refractory to multiple lines of different therapies were treated with a single infusion of tisagenlecleucel at a median dose level of 3.1 million viable CAR-T cells per kilogram [3]. Within the following 3 months, 81% of the patients achieved minimal residual disease (MRD)-negative complete remission. At 12 months post therapy, the event-free survival was 50% and the overall survival rate was 76%. Based on those impressive results, tisagenlecleucel (Kymriah, Novartis) was granted FDA and EMA approval for the therapy of r/r ALL, setting a milestone in the history of T-cell therapy as the first officially approved CAR-T-cell product. Nevertheless, relapsing ALL-blasts with down-regulated or abrogated CD19 expression remain a critical obstacle to maintaining permanent remissions [7]. Thus, back-up antigens, such as CD22, have garnered increasing attention. In a recent clinical trial, CD22-CAR-T cells displayed the potential to mediate complete remissions in patients relapsing with CD19-negative blasts [71]. The medium remission duration, however, was only 6 months and could be attributed to the diminished density of CD22 on relapsing cancer cells [71]. 

In contrast to ALL, where complete remissions are frequent and dealing with relapse poses a major challenge, CAR-T-cell therapy in chronic lymphocytic leukemia (CLL) is characterized by infrequent but lasting complete remission rates [72]. A variety of target antigens encompassing CD20 [73], CD23 [74], ROR1 [75], κ-light chain [36], and FuR [76] were investigated in CLL. Equally to ALL, the most promising clinical data were obtained using CD19-CAR-T cells [77]. Nonetheless, the overall complete remission rate of CD19-CAR-T cells in CLL is only 29% [77]. Acquired proliferative and metabolic dysfunctionalities evident in T cells from CLL patients might contribute to the lower efficacy of CAR-T-cell therapy in CLL [78]. 

Contrary to ALL and CLL, clinical data concerning CAR-T cells in acute myeloid leukemia (AML) still require maturing. Early data from a trial investigating CAR-T cells targeting the carbohydrate antigen Lewis-Y on myeloid blasts provided proof-of-principle data for CAR-T-cell efficacy in AML, but no lasting remission could be reported [79]. Owing to universal presence on myeloid blasts at primary diagnosis and at relapse, CD33 and CD123 have emerged as the frontrunner target antigens receiving the most attention in current clinical trials [80]. Especially, CAR-T cells targeting CD123, which is expressed on leukemia-initiating-cells [81], could induce complete remissions in an ongoing trial (NCT02159495) led by the City of Hope National Medical Center [80]. Another promising target antigen is FMS-like tyrosine kinase 3 (FLT3), which could be successfully exploited using FLT3-targeting CAR-T cells demonstrating potent reactivity against AML blasts expressing wild-type or FLT3 with internal tandem duplication (FLT3-ITD) [82]. In the following we will discuss the merits and challenges of CSPG4 as a target for the CAR-T-cell therapy of leukemia (Figure 3).

#### 2.2.1. CSPG4-CAR-T Cells: Merits (Leukemia)

The majority of AML target antigens, such as CD123, CD33, and FLT3, can be detected on the surface of hematopoietic stem cells (HSCs) [81,82]. Hence, in AML therapy, a pivotal predicament is posed by myeloablation resulting from on-target/off-leukemia activity. Investigators from Pennsylvania observed that CAR-T cells directed against CD123 eliminated primary AML in immunodeficient mice, but at the expense of normal HSCs, which were also completely erased by CD123-CAR-T cells resulting in permanent pancytopenia [83]. Thus, the use of CAR-T cells targeting CD123 may necessitate a subsequent rescue allogeneic bone marrow transplant, restoring hematopoiesis but simultaneously shutting down CAR-T-cell activity during conditioning chemotherapy and total body radiation. Given the fact that successful CAR-T-cell applications correlate with extended persistence of engineered T cells, concerns of leukemia relapse after bone marrow transplantation are stirred [84]. Equally, potent pre-clinical activity of CD33-specific CAR-T cells displayed in a xenograft leukemia model coincided with pronounced cytopenias emanating from on-target/off-leukemia toxicity against myeloid progenitors [85]. To reconcile strong anti-leukemia with sustained hematopoiesis after CD33-CAR-T-cell transfer, researchers surrounding Saar Gill devised a strategy to combine CD33-CAR-T-cell application with an allogeneic bone marrow transplant using CRISPR-modified HSCs that do not express CD33 anymore [86]. Nevertheless, genetic manipulation of stem cells raises concerns about a potential malignant transformation of HSCs by the inadvertent knockout of tumor suppressor genes. In aggregate, a possible merit connoted with CSPG4-CAR-T cells in AML is the exertion of anti-leukemia activity against CSPG4-positive blasts without affecting HSCs, which were found to be universally CSPG4-negative [87]. Conspicuously, CSPG4 expression is preponderant in the dismally-fated mixed lineage leukemias (MLL), which comprise around 10% of all leukemias and are characterized by specific translocations involving 11q23 [16]. The mechanisms responsible for CSPG4-presence in MLL have not been elucidated so far, but some hypotheses indicate that CSPG4 might be up-regulated ensuing promotor demethylation caused by the translocation-induced disruption of the MLL1 methyltransferase gene. In early 2019, we could provide the first proof-of-principle data on CSPG4-CAR-T-cell efficacy against MLL-leukemia using KOPN8 B-cell precursor leukemia cells as a model [88]. Against the backdrop of reports from two recent clinical trials about B-ALL cells undergoing a switch from the lymphoid lineage towards a myeloid phenotype to abrogate CD19 expression and evade destruction by CD19-CAR-T cells, the quest for non-lineage-restricted backup antigens has gained attraction [89]. Hypothetically, the capacity to induce lineage switches as a means of immune evasion may be more frequent in MLL-leukemias, which are characterized by an inherent plasticity derived from the general disturbance in epigenetic regulation [90]. Physiologically, CSPG4 is neither expressed in the lymphoid nor the myeloid branch of hematopoiesis [16]. Therefore, CSPG4 presence might be preserved in MLL irrespective of lineage switches. Moreover, CSPG4 has been implicated in promoting central nervous system infiltration and leukemia invasiveness [91]. Finally, CSPG4 is supposed to confer protection against chemotherapy on MLL blasts by preventing leukemia cells to egress from the bone marrow, which constitutes a chemoprotective environment maintained by bone marrow stroma cells [92]. Therefore, shutting down CSPG4 expression on MLL cells to evade CAR-T-cell therapy may reduce invasive capacities and enhance chemosensitivity.

#### 2.2.2. CSPG4-CAR-T Cells: Challenges (Leukemia)

Besides the issues with potential on-target/off-tumor activities, which are discussed in detail in the melanoma section, the heterogenous expression of CSPG4 on leukemia cells remains a critical challenge. Although CSPG4 presence has been confirmed in lymphoid and myeloid leukemias, CSPG4 seems to be primarily expressed in 11q23 rearranged MLL [16]. In a cohort comprising 166 patients with AML, strong CSPG4 expression (>25% of blasts) was only detected in 18 patients (11%), which were found to bear 11q23 abnormalities [87]. In another study, 13 of 37 patients with AML (35.1%) suffered from CSPG4-positive disease, all of which displayed MLL rearrangement [37]. Of note, CSPG4 staining in those two studies was carried out using the 7.1 anti-CSPG4 monoclonal antibody. Given the diverse glycosylation patterns of CSPG4, studies probing CSPG4 expression for CAR-T-cell therapy should rely on antibodies, which are not significantly influenced by the glycosylation status of CSPG4, such as the anti-CSPG4 antibody 9.2.27. To date, no comprehensive studies analyzing CSPG4 expression on leukemia cells using the 9.2.27 antibody exist, providing an avenue for future research. Collectively, the limited expression profile of CSPG4 on leukemia cells indicates that CSPG4-CAR-T cells are primarily conceivable as part of a combinatorial targeting approach to MLL. Moreover, CSPG4 expression should be monitored for delayed up-regulation at different time-points during leukemia therapy, especially in the case of phenotypic changes or alterations in leukemia biology, such as increased invasiveness. In contrast to melanoma, where CSPG4 acts as an oncogenic driver by providing growth-promoting signals and blocking apoptosis (see melanoma section), CSPG4 up-regulation in MLL leukemia cells is perceived to be a side-product resulting from the deregulated epigenetic regulation inherent to this leukemia subtype. Consequently, CSPG4 shut down is not associated with overt repercussions on leukemia progression, predisposing the emergence of CSPG4-negative escape variants. 

### 2.3. Glioblastoma

Given a lack of curative options rendering long-term survival a rare event, CAR-T-cell therapy for high-grade gliomas has incited a lot of enthusiasm reflected by a steadily expanding clinical trial landscape evaluating the efficacy of CAR-T cells against glioblastoma (GBM). Encouraged by initial studies proving the safe application of engineered first generation CAR-T cells targeting IL-13Rα2 [93], which is co-expressed on bulk GBM cells and glioma stem cells [94], investigators from the City of Hope have registered a follow-up trial to analyze the efficacy of 4-1BB-co-stimulated second-generation IL-13Rα2-specific CAR-T cells in relapsing and refractory GBM (NCT02208362, NCT00730613). With the trial still ongoing, a dramatic clinical response achieved in a 50-yeard old patient with recurrent GBM was shared in 2016. After 6 intracavitary and 10 intraventricular infusions of IL-13Rα2-specific CAR-T cells, a striking complete remission of all lesions could be observed lasting 7.5 months [95]. This is the first report to demonstrate the capacity of CAR-T cells to induce complete tumor regressions in GBM. 

A considerable number of primary GBM harbor genetic alterations in the epidermal growth factor receptor gene resulted in the de-novo generation of a unique truncated EGFR variant, termed EGFR variant III (EGFRvIII) [96]. Several clinical trials with EGFRvIII-specific CAR-T cells were performed or are ongoing (NCT02331693. NCT02873390, NCT02664363, and NCT03170141). In a dose escalation study conducted at the NCI using third generation (CD28 and 4-1BB co-stimulatory domains) EGFRvIII-specific CAR-T cells, which were co-administered with IL-2 by intravenous infusion following lymphodepleting chemotherapy, no objective responses could be documented [97]. The limited persistence of CAR-T cells after infusion might provide an explanation for the absent therapeutic efficacy. Unfortunately, one patient treated with the maximum dose level died from pulmonary toxicity arising within hours after CAR-T-cell infusion. Another trial (NCT02209376) examining EGFRvIII-CAR-T cells found that CAR-T cells are capable of infiltrating tumor sites, but are functionally inhibited by a protective TME, the immunosuppressive power of which positively correlated with CAR-T-cell doses [98]. Furthermore, EGFRvIII was found to be heterogeneously expressed on glioma cells and EGFRvIII showed progressive down-regulation in response to CAR-T cells infusion [98]. To tackle the heterogeneity of EGFRvIII expression and to counteract the outgrowth of EGFRvIII-negative tumor cells, CART.BITE T cells have been recently developed, which are equipped with a CAR specific for EGFRvIII and a bi-specific T-cell engager (BiTE) directed to EGFR [99]. These CART.BITE T cells exhibited control of tumors with heterogenous EGFRvIII expression without inducing any serious toxicity [99]. 

Data gathered at Baylor College of Medicine demonstrated the safe clinical application of second-generation CAR-T cells redirected to HER2 (NCT01109095). Regarding therapeutic power, the best response was a single partial remission followed by several patients with stable disease. No significant expansion of CAR-T cells could be observed, but CAR-T-cell presence could be monitored for up to 1 year. Other trials investigating HER2-specific CAR-T cells are ongoing (NCT03389230, NCT02442297, and NCT02713984). 

Finally, several other antigens on glioblastoma are targeted in other trials, e.g., ephrin type-A receptor 2 (EphA2) (NCT02575261), GD2 (NCT03252171), and mucin 1 (MUC1) (NCT02839954, NCT02617134). In the following we will discuss the merits and challenges of CSPG4 as a target for the CAR-T-cell therapy of glioblastoma (Figure 4).

#### 2.3.1. CSPG4-CAR-T Cells: Merits (Glioblastoma)

CSPG4 is as a marker for glioblastoma stem cells (GSCs), which are instrumental in GBM progression [58]. Additionally, due to their intrinsic self-renewal capacity, GSCs are also closely associated with GBM recurrence [100]. Hence, GSC eradication constitutes a vital prerequisite for durable remissions. Beard et al. demonstrated the successful killing of several GSC cell lines derived from primary tumors using CSPG4-CAR-T cells [58]. Moreover, CSPG4 executes distinct functions in GSCs, such as enhancing responsiveness to PDGF [29]. Consequently, shutting down CSPG4 expression to escape destruction by CSPG4-CAR-T cells might incur functional impairments on GSCs and compromise GBM progression. Besides, high-grade gliomas such as anaplastic glioma and GBM are characterized by extensive neovascularization [101]. Strikingly, CSPG4 was particularly enriched in areas with high proliferation, indicating that CSPG4 might be especially up-regulated in progressive areas [102]. This assumption is backed by data showing that CSPG4 overexpression promotes GBM growth, proliferation and neovascularization [29]. Mechanistically, CSPG4 expedites the formation of tumor blood vessels by binding and sequestering angiostatin, which is an inhibitor of angiogenesis [102]. In sum, CSPG4-CAR-T cells could exert a dual hit on GBM by concomitantly targeting GSCs and tumor-associated vasculature. Similar to melanoma and contrary to leukemia, CSPG4 acts as an oncogenic driver delivering growth promotion and survival signals to GBM cells. This is exemplified by a recent study investigating the impact of blocking CSPG4 activity in temodal-resistant GBM cells using the diene-type sesquiterpene furanodienone [103]. The inhibition of CSPG4 signaling diminished AKT and ERK activity, reduced tumor-associated inflammation, decreased the transcription of pro-mitogenic early growth response protein 1 (EGR1), and facilitated the execution of pro-apoptotic programs [103]. Given the pro-oncogenic features mediated by CSPG4, antigen-loss during CSPG4-CAR-T cell therapy will be rendered less likely. Pellegatta et al. detected a high uniform expression of CSPG4 in 67% of a cohort of 46 GBM specimen, with all but one of the remaining samples showing CSPG4 positivity, albeit not as high as the others [104]. Besides, intracranial administration of CSPG4-CAR-T cells effectively controlled GBM growth in clinically relevant xenograft mouse models without obvious immune evasion emanating from CSPG4-neagtive GBM cells [104]. Surprisingly, GBM cells relapsing 1 week after intracranial inoculation displayed a drastic increase in CSPG4 expression (99% vs. 49%) [104]. The reason for augmented CSPG4 expression was attributed to TNF release from microglial cells residing in the pro-inflammatory TME. This resulted in a subsequent TNF-mediated activation of the NFκB pathway, promoting the up-regulation of CSPG4 on GBM cells [104]. Importantly, TNF production represents a canonical effector function of CAR-T cells. Thus, the TNF-inducible expression of CSPG4 in combination with TNF elaborated from CSPG4-CAR-T cells may create a feed-forward loop culminating in uniform CSPG4 expression on GBM cells, paving the way for complete tumor eradication. Remarkably, no comparable changes in expression levels have been observed regarding HER2 and IL-13Rα2 [104]. Lastly, elevated resistance of GBM cells to radiation therapy and chemotherapeutic agents has also been associated with CSPG4 up-regulation. GBM cells expressing CSPG4 displayed resistance to ionizing radiation via swift recognition of DNA damage and subsequent cell cycle arrest at G1 and G2 checkpoints preventing uncontrolled proliferation and accumulation of cytotoxic chromosomal damage [105]. Additionally, CSPG4 effectuates chemoresistance to a variety of drugs including the first line agent temodal, by activation of integrin-dependent PI3K/AKT signaling [27]. In summary, antigen-loss of CSPG4 in response to CSPG4-CAR-T-cell therapy might re-sensitize GBM cells to radiation therapy and chemotherapeutic agents.

#### 2.3.2. CSPG4-CAR-T Cells: Challenges (Glioblastoma)

Apart from general safety issues arising from potential on-target/off tumor toxicities which are elaborated on in the melanoma section, GBM poses some distinct challenges to CSPG4-CAR-T-cell therapy. First, CSPG4 is abundantly expressed on GBM cells but not universally on every cell [104]. Therefore, to counteract the emergence of antigen-negative clones, strategies combining CSPG4-CAR-T cells with CAR-T cells targeting IL-13α2R, EGFRvIII or HER2/neu are worthwhile exploring. Moreover, the TNF-mediated up-regulation of CSPG4 described above plays an important role in augmenting CSPG4 antigen density on GBM cells [104]. Nevertheless, the application of CSPG4-CAR-T cells still awaits its first human trial to provide valid clinical data on safety. Without those data, possible safety concerns are stirred by a potential TNF-mediated up-regulation of CSPG4 on non-malignant host cells [104]. The subsequent attack of CSPG4-CAR-T cells could culminate in serious damage to the central nervous system. The severity of T-cell-mediated brain toxicity can be judged from a clinical trial evaluating autologous anti-MAGE-A3-TCR-modified T cells in cancer patients [106]. Due to an unexpected expression of MAGE-A12 in the brain, which triggered cross-reactive activity by MAGE-A3-specific engineered T cells, two patients contracted lethal necrotizing leukoencephalopathy and periventricular leukomalacia following 1–2 days post CAR-T-cell infusion [106]. To minimize the risk of permanent damage inflicted upon non-malignant brain cells by CSPG4-CAR-T cells in the wake of a potential TNF-mediated CSPG4 up-regulation on host cells, the treatment regime could be initiated by several cycles of transiently modified RNA-CAR-T cells. In case of absent brain toxicity, clinicians could proceed to infuse genetically-engineered CAR-T cells. Finally, targeting CSPG4-positive GMB-associated vasculature could result in acute cranial hemorrhage, with potentially lethal consequences. Clinical data obtained from GBM patients receiving the anti-angiogenetic agent bevacizumab did not reveal a significant risk of bleeding [38]. However, owing to the different functional properties and potencies of CAR-T cells and monoclonal antibodies, direct comparisons and extrapolations are difficult. Thus, patients undergoing CSPG4-CAR-T-cell therapy should be thoroughly examined for signs of intracranial hemorrhage.

### 2.4. Triple-Negative Breast Cancer (TNBC)

The absent expression of estrogen receptors (ER), progesterone receptors, and epidermal growth factor receptors (EGFR) coupled with an insufficient response to chemotherapy renders triple-negative breast cancer (TNBC) frequently resistant to standard breast cancer therapies [107]. Hence, novel therapeutic modalities, such as the use of CAR-T cells, have attracted attention. Recently, Zhou et al. engineered T cells with a CAR derived from the TAB004 monoclonal antibody, which specifically binds to an aberrant glycoform of MUC1, termed tMUC1 [108]. T cells transfected with tMUC1-specific CARs exhibited antigen-specific cytotoxicity against several TNBC cell lines. Moreover, a single application of tMUC-1 CAR-T cells could significantly impede tumor growth in a TNBC xenograft model. Nevertheless, progressive disease, most probably owing to pronounced T-cell exhaustion, could be observed in the long run. Importantly, tMUC1 is present in greater than 95% of all TNBC, while no significant tMUC1 staining could be detected on normal breast epithelium [108]. Another aberrant glycoform of MUC1, TnMUC1, has also been exploited as a CAR-T-cell target for TNBC [109]. Based on the TnMUC1 antibody E5E, investigators from the University of Pennsylvania designed a CAR-construct, which capacitated T cells to specifically recognize TNBC cells [109]. In sum, deploying CAR-T cells redirected to MUC1 glycoforms in TNBC therapy represents a promising approach still awaiting corroboration with clinical data. So far two active clinical trials evaluating the targetability of MUC1 derivatives have been registered and are about to commence recruitment (NCT04020575, NCT04025216). 

Tchou et al. focused on mesothelin as a potential target structure in TNBC. They found mesothelin to be overexpressed in 67% of TNBC samples, with limited expression on other breast cancer subtypes and absent expression in normal breast epithelial cells [110]. Moreover, CAR-T cells specific for mesothelin evinced cytotoxicity towards TNBC cells encouraging further investigation of this antigen in clinical trials [110] (NCT02580747). Of note, mesothelin-positive TNBC are associated with a lower overall and disease-free survival, are more prone to spawn metastases, and are more abundant in elderly patients [111]. 

Motivated by the broad but variable expression of NKG2D ligands on TNBC cells, Han et al. fused the backbone of a conventional second-generation CAR to the extracellular part of NKG2D [112]. CAR-T cells expressing this NKG2D-CAR recognized TNBC cells in vitro and significantly delayed tumor growth in NSG mice subcutaneously inoculated with TNBC cells [112]. Currently, a phase 1 clinical trial has been launched by Celyad to assess the safety and clinical efficacy of repetitive injections of T cells equipped with a NKG2D-CAR targeting NKG2D-ligands on a variety of different cancer entities including TNBC. 

Furthermore, the arsenal of target structures for the CAR-T cell therapy of TNBC encompasses the integrin αvβ3 [42], the receptor tyrosine kinase AXL [113], the integrin-like cell surface protein TEM8 [114], the receptor tyrosine kinase c-MET [115](NCT01837602), and ROR1 [116]. In an ongoing phase 1 clinical trial, TNBC patients are exposed to incrementing doses of ROR1-specific CAR-T cells (3.3 × 10^5^, 1 × 10^6^, 3.3 × 10^6^ and 1 × 10^7^ cells/kg) (NCT02706392). Preliminary results updated in 2018 indicated that the transfer of ROR1-CAR-T cells is not associated with any serious toxicities [117]. Moreover, those CAR-T cells were found to expand and infiltrate tumor tissue [117]. In the following we will discuss the merits and challenges of CSPG4 as a target for the CAR-T-cell therapy of TNBC (Figure 5).

#### 2.4.1. CSPG4-CAR-T Cells: Merits (TNBC)

Using the monoclonal antibody 225.28, CSPG4 could be identified in 32 of 44 (72.7%) primary TNBC lesions as well as in pleural effusions from 12 patients with TNBC [118]. Strikingly, CSPG4 expression exhibited a clear preponderance to TNBC, with few ER^+^ or Her2^+^ (EGFR2) breast cancer subtypes expressing CSPG4 [118]. Additional analyses of immunohistochemical staining from 240 breast cancer specimens conducted by Hsu et al. corroborated the presence of CSPG4 in TNBC, but contrary to previous reports, CSPG4 expression on breast cancer cells was not confined to the TNBC subtype [119]. Moreover, the degree of CSPG4 expression was inversely correlated with overall survival and time to recurrence [119]. A possible explanation for the dismal prognosis associated with CSPG4 up-regulation is provided by findings implicating CSPG4 in distant metastasis formation. Mechanistically, interactions between chondroitin sulfate side chains of CSPG4 with P-selectin are thought to result in tumor cell activation and augmented survival of circulating breast cancer cells. The adhesion molecule P-Selectin evinces elevated expression on breast cancer cells and can be triggered by CSPG4 on the surface of cancer cells or by CSPG4-expressing stromal cells residing in the tumor microenvironment (TME) [21]. Notably, the removal of surface chondroitin sulfate glycosaminoglycans via chondroitinase treatment significantly diminished lung metastases in a murine breast cancer model [21]. Hence, counteracting CSPG4-CAR-T cells in TNBC via down-regulating CSPG4 might impair metastasis formation and stunt breast cancer progression. Besides, CSPG4 presence was also detected in TNBC cancer stem cells, which are regarded as a major source for relapse and resistance [118]. Another major benefit of CSPG4-CAR-T cells is posed by the capacity to concomitantly target CSPG4-positive TNBC cells and cancer-associated fibroblasts, which assume a crucial role in maintaining the tumor microenvironment [120]. Additionally, a special subpopulation of stromal cells characterized by Nestin, CSPG4, α-smooth muscle actin, and PDGFR-α expression are recruited to the TME by breast cancer cells [121]. Those CSPG4-positive stromal cells drive the invasion and anchorage-independent growth of cancer cells by supplying the pro-inflammatory cytokine IL-8 [121]. In sum, CSPG4-expressing cancer-associated stromal cells represent an important source for growth and proliferation stimuli. Further experiments evaluating the effects of CSPG4 ablation on tumor growth in a murine breast cancer model revealed a reduced early progression phase [122]. This finding was rationalized by a disturbance of tumor-associated vasculature resulting in hypoxia and nutrient deprivation [122]. In aggregate, CSPG4-CAR-T cells have the potential to mount a concerted attack against various targets, including primary TNBC cells, stromal cells, and tumor surrounding blood vessels. Moreover, the complex role of CSPG4 in the progression of TNBC and the architecture of the TME renders swift disposal of CSPG4 more difficult.

#### 2.4.2. CSPG4-CAR-T Cells: Challenges (TNBC)

Antigen dependence has emerged as the central Achilles heel of CAR-T-cell therapy, which is evident in the selective survival and outgrowth of antigen-negative tumor cells [7]. In TNBC, CSPG4 displays a broad, but not uniform expression [118]. Hence, deploying CSPG4-CAR-T cells as monotherapy bears risks of sparing CSPG4-negative tumor cells, which could provide the basis for refractory disease or relapse. In order to tackle this issue, a multi-hit approach comprising CSPG4-CAR-T cells and CAR-T cells directed against other TNBC antigens seems reasonable. Especially, the combination of CSPG4-CAR-T cells with CAR-T cells redirected to tMUC1 appears promising, even though no data analyzing this combination have been obtained so far. The antigen tMUC1 is uniformly expressed on TNBC cells [108], but owing to its nature as an aberrant glycosylation product, it is neither an oncogenic driver nor instrumental in tumor progression, which facilitates tMUC1 shutdown. On the flipside, CSPG4 is less uniformly expressed than tMUC1, but exerts important functions in metastasis formation and in the TME, qualifying CSPG4 as an oncogenic driver in TNBC [21,121]. In summary, CSPG4-CAR-T cells and tMUC1-CAR-T cells evince complimentary characteristics, and studies elucidating possible synergistic effects derived from simultaneously targeting CSPG4 and tMUC1 in TNBC are warranted. Another major challenge to CAR-T-cell approaches in general is posed by secondary antigen-loss during CAR-T-cell therapy. Mechanisms for antigen shutdown include mutational loss and epigenetic silencing, among others [7]. A key feature of TNBC biology is a high degree of genetic instability, which is in part expedited by hereditary or acquired mutations in genes, e.g., breast cancer type 1 susceptibility protein (BRCA1), encoding proteins responsible for orchestrating DNA damage repair and cell cycle control [123]. As a consequence of this genetic instability, TNBC displays a steady accumulation of mutations and other genetic alterations [123]. Speculatively, this might be beneficial in disposing of target antigens, such as CSPG4, to escape antigen-specific therapies. To mitigate these concerns, CSPG4-CAR-T cells could be co-administered with drugs directed against the compromised DNA repair machinery of certain TNBC cells, such as the FDA-approved inhibitor of poly(ADP-ribose) polymerase (PARP), olaparib. Hypothetically, this could counteract mutational target loss by eradicating TNBC cells with pronounced genetic instability and DNA damage. Finally, concerns raised by potential on-target/off-tumor toxicities, which are discussed in detail in the melanoma section, also apply to CSPG4-CAR-T-cell therapy of TNBC. With the central nervous system being a primary site of metastasis in breast cancer, CSPG4-CAR-T cells are expected to enter the brain, infiltrate metastatic lesions and mediate effector functions, such as cytokine secretion and target cell lysis. Various reports have described CSPG4 expression in pericytes surrounding tumor-associated endothelial cells [122,124]. In patients with a high metastatic burden in the central nervous system, the use of CSPG4-CAR-T cells might entail the risk of causing acute cerebral hemorrhage by eliminating tumor-associated vasculature. 

## 3. Conclusions

Recent approval of CD19-CAR-T cells for use in patients with certain lymphomas and acute lymphoblastic leukemia by US-American and European regulatory agencies has transformed CAR-T-cell therapy from a mere experimental therapeutic modality to a licensed living drug. Having reached this milestone, current efforts are concentrated on dealing with the relapse of antigen-negative cancer cells in leukemia and on achieving significant tumor regressions in dismally-fated solid malignancies, such as melanoma, glioblastoma, and triple-negative breast cancer. In this process, a major focus is placed on enlarging the arsenal of targetable antigens. The proteoglycan CSPG4 has been found on the surface of melanoma cells, MLL-rearranged leukemia cells, glioblastoma cells, and triple-negative breast cancer. Hence, analyzing the performance of CSPG4-CAR-T-cell therapy in those malignancies has garnered interest. In melanoma, CSPG4 acts as an oncogenic driver, supporting growth and survival of melanoma cells, which reduces the risk of antigen-loss. In leukemia, CSPG4 expression is confined to the ill-fated MLL-rearranged subtype, which is characterized by a high intrinsic plasticity facilitating antigen-loss. Thus, CSPG4-CAR-T cells might be useful in combinational approaches incorporating other CAR-T cells with different specificities. Similarly to melanoma, CSPG4 exerts pro-oncogenic functions in GBM, fostering tumor growth. Moreover, CSPG4 is expressed on GBM stem cells and on tumor-associated vasculature. Consequently, CSPG4-CAR-T cells could mount a multi-hit attack against GBM cells, simultaneously engaging bulk tumor cells, tumor stem cells and tumor blood vessels. In triple-negative breast cancer, CSPG4 has been associated with metastasis formation and crucial functions in the tumor microenvironment. Therefore, CSPG4-CAR-T cells could launch a dual attack on bulk breast cancer cells and on the tumor-protective TME. Despite all those beneficial attributes, the use of CSPG4-CAR-T cells still poses challenges to consider. A major caveat is presented by the potential on-target/off-tumor activity against non-neoplastic pericytes, which could incur serious bleeding. Besides, CSPG4 was not shown to be associated with any pro-oncogenic function in leukemia, which predisposes the generation of CSPG4-negative clones. In sum, judging from the preclinical data obtained so far on CSPG4 and the couple of preclinical studies evaluating CSPG4-CAR-T cells, CSPG4 appears to be an attractive antigen which should be evaluated in clinical trials to elucidate whether the merits or challenges associated with CSPG4-CAR-T cells prevail in clinical settings.

## Figures and Tables

**Figure 1 ijms-20-05942-f001:**
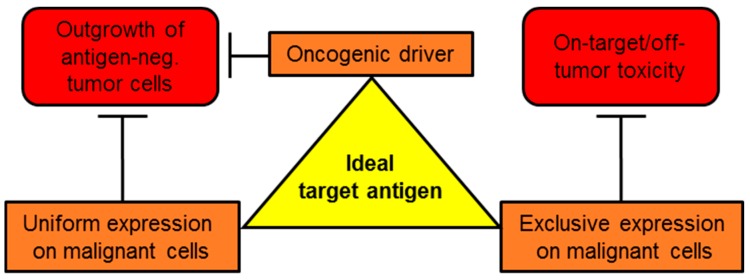
Schematic representation featuring the attributes of an ideal target antigen for chimeric antigen receptor (CAR)-T-cell therapy in general. See text for details.

**Figure 2 ijms-20-05942-f002:**
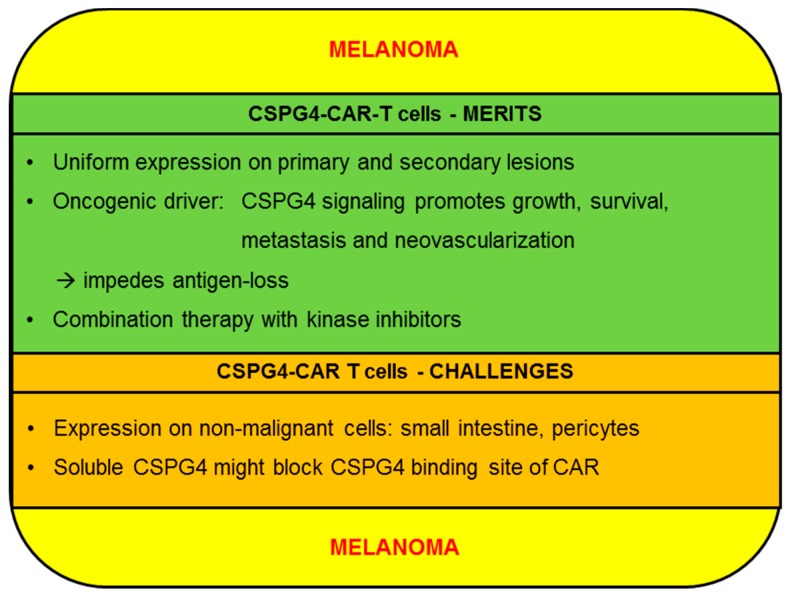
Schematic overview of the merits and challenges of chondroitin sulfate proteoglycan 4 (CSPG4) for the CAR-T-cell therapy of melanoma.

**Figure 3 ijms-20-05942-f003:**
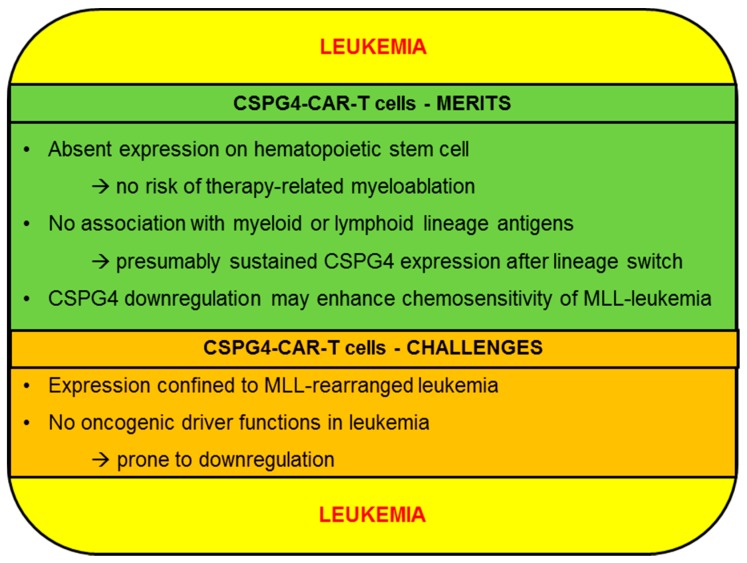
Schematic overview of the merits and challenges of CSPG4 for the CAR-T-cell therapy of leukemia.

**Figure 4 ijms-20-05942-f004:**
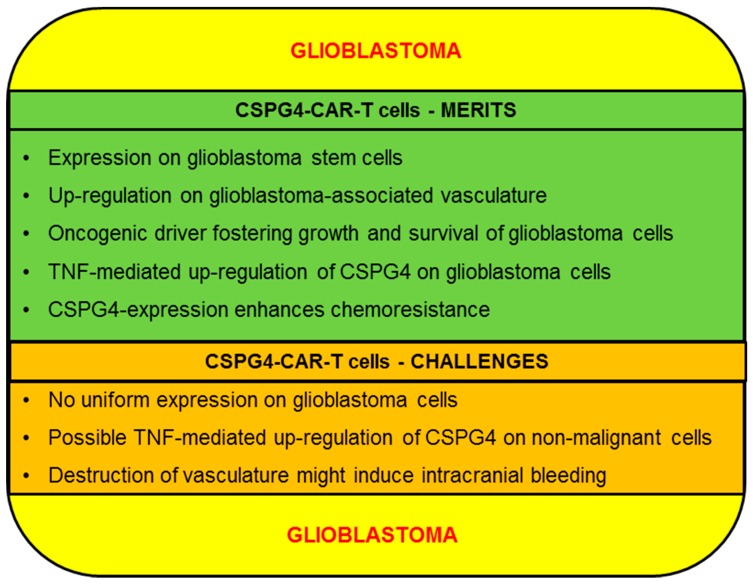
Schematic overview of the merits and challenges of CSPG4 for the CAR-T-cell therapy of glioblastoma.

**Figure 5 ijms-20-05942-f005:**
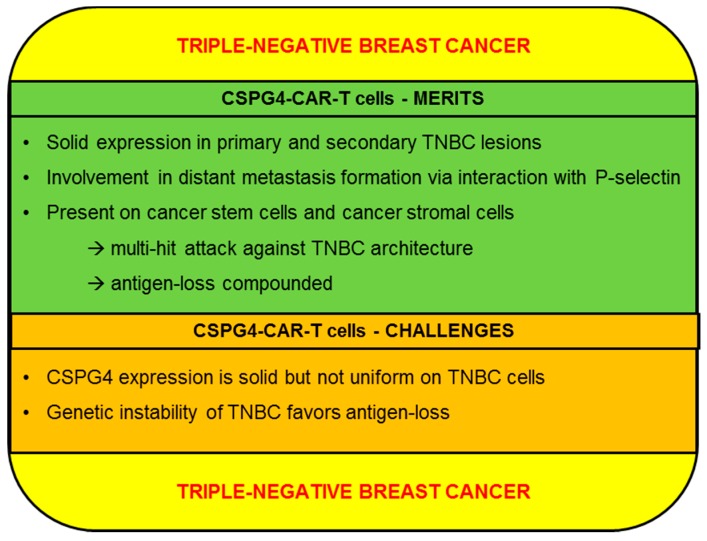
Schematic overview of the merits and challenges of CSPG4 for the CAR-T-cell therapy of triple-negative breast cancer.

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
