# Peer review of "CSPG4 as Target for CAR-T-Cell Therapy of Various Tumor Entities–Merits and Challenges"

_ijms, 2019, doi:10.3390/ijms20235942_

Round 1

Reviewer 1 Report

This is an excellent review of the merits and challenges of CSPG4 as a target antigen in CAR-T cell therapy, focusing on melanoma, leukemia (especially MLL), GBM and TNBC as tumors that show elevated expression of surface CSPG4.

Following broad introduction on CAR-T cell therapy, the author present a detailed description of the structure and function of CSPG4 and its speculated roles in tumorigenicity and metastasis. The authors then continue with a comprehensive review of clinical and preclinical data of CAR-T cell therapy of each of the four types of cancer, each accompanied by thorough analysis of the merits and challenges of targeting CSPG4 by CAR-T cells.

Major comment:

While discussing on-target/off-tumor toxicity the authors refer to transient CAR expression achieved by mRNA electroporation, but do not mention other potential strategies for counteracting such toxicity, such as the use of inhibitory CARs or combinational antigen recognition (achieved, for example, by ‘split’ recognition, SynNotch-induced expression, etc.). The authors are requested to refer to these approaches and suggest potential target antigens, which could improve selectivity of treatment.

Minor comments:

Line 39 – please add abbreviation for scFv Line 43 - should be immunoglobulin Lines 106-107 –please add here references for expression in other tumor types

Reviewer 2 Report

This review article summarizes merits and challenges of CSPG4 as a target for CAR-T cell therapy in different cancer types. CSPG4 is a potential target due to its low tissue distribution in normal tissues and overexpression in tumors. It is a well written and thorough review of CSPG4 CAR-T cells for treatment of different cancer types. To further enhance the understanding and the challenges faced by CSPG4 CAR-T cells please address the following :

Effects of cytokine associated with Tumor Microenvironment (TME) on CAR-T cells in general or preferably on CSPG4 CAR-T cells (if studies are available). Effects of immunosuppressive cell types (TAMs, MDSCs, Tregs etc) on the CSPG4 CAR-T cells in the TME and their associated signaling and potential targets. Present a table of clinical trials involving the CSPG4 CAR-T cells (with and without other combination therapy) in different cancer types and their outcomes. Brief background about the CSPG4 distribution in normal tissues and their normal function. The introduction about CSPG4 protein can be broken down into sub-topics such as 1.CSPG4 in normal tissue distribution and their up-regulation causes in cancer development, 2. CSPG4 structure and function, 3. changes in signaling in diseased conditions etc.
